# Recent Breakthroughs in Using Quantum Dots for Cancer Imaging and Drug Delivery Purposes

**DOI:** 10.3390/nano13182566

**Published:** 2023-09-15

**Authors:** Aisha Hamidu, William G. Pitt, Ghaleb A. Husseini

**Affiliations:** 1Biomedical Engineering Program, College of Engineering, American University of Sharjah, Sharjah P.O. Box 26666, United Arab Emirates; g00087960@alumni.aus.edu; 2Department of Chemical Engineering, Brigham Young University, Provo, UT 84602, USA; pitt@byu.edu; 3Materials Science and Engineering Program, College of Arts and Sciences, American University of Sharjah, Sharjah P.O. Box 26666, United Arab Emirates; 4Department of Chemical and Biological Engineering, College of Engineering, American University of Sharjah, Sharjah P.O. Box 26666, United Arab Emirates

**Keywords:** quantum dots, functionalization, in vitro imaging, in vivo imaging, drug delivery

## Abstract

Cancer is one of the leading causes of death worldwide. Because each person’s cancer may be unique, diagnosing and treating cancer is challenging. Advances in nanomedicine have made it possible to detect tumors and quickly investigate tumor cells at a cellular level in contrast to prior diagnostic techniques. Quantum dots (QDs) are functional nanoparticles reported to be useful for diagnosis. QDs are semiconducting tiny nanocrystals, 2–10 nm in diameter, with exceptional and useful optoelectronic properties that can be tailored to sensitively report on their environment. This review highlights these exceptional semiconducting QDs and their properties and synthesis methods when used in cancer diagnostics. The conjugation of reporting or binding molecules to the QD surface is discussed. This review summarizes the most recent advances in using QDs for in vitro imaging, in vivo imaging, and targeted drug delivery platforms in cancer applications.

## 1. Introduction

Cancer is a group of diseases characterized by the rapid growth of abnormal cells within the body. In most cancer cases, the mutations or changes in the expression of proto-oncogenes, tumor suppressor genes, and DNA repair genes are responsible for cancer development [1]. The majority of cancers are attributed to genetic (mutations, hormones, immune conditions) or environmental (radiation, chemicals, pollutants) factors, in addition to indicators of an unhealthy lifestyle (poor diet, tobacco smoking) [2,3]. Furthermore, the risk of cancer increases significantly with increasing age.

Cancer is one of the leading causes of death worldwide. According to the World Health Organisation (WHO), the number of cancer deaths was nearly 10 million in 2020 [4,5,6]. The number of new cases is estimated to be 28.4 million by 2040 [7]. The fight against cancer remains one of the most significant issues facing the world. Current conventional means to battle cancer have significant drawbacks, including but not limited to toxicity and non-specificity of conventional chemotherapeutics [8]. Early detection and intervention have a significant positive impact on patient outcomes.

In recent decades, research into and applications of nanomedicine have grown significantly, especially in cancer diseases [9,10,11,12,13,14]. Such research has shown great potential to overcome previous challenges relating to early tumor detection, accurate diagnoses, and individualized treatment [15,16,17]. The primary benefit of nanomedicine in cancer therapy is the tiny size of nanoparticles, which allows them to function at the molecular level, thereby enhancing diagnosis and improving the chances of achieving innovative targeting strategies at the molecular level [18,19,20,21,22]. For example, some nanoparticles work by binding to cancer biomarkers such as circulating tumor cells, circulating tumor DNA, exosomes, and specific cancer-associated proteins [23,24,25].

Nanometer-scale materials (1–100 nm) display intriguing properties due to their small size [26]. These novel properties are often due to the quantum confinement and surface effects affected by their small size [27]. The quantum confinement effect confines moving electrons within a small volume, producing unique optical and electronic effects. As for surface effects, the chemical reactivity of the surface usually increases as the size decreases, while the melting point usually decreases [28,29]. These novel optical and thermal properties of nanomaterials can be useful for both in vivo and in vitro applications via the active interaction with molecular components at the cellular level.

Several nanoparticles [30,31,32,33] have been investigated for cancer diagnosis and therapy. Nowadays, quantum dots (QDs), often referred to as “artificial atoms”, are a hot topic in cancer nanomedicine. They were first described in 1981 by Alexey Ekimov [34]. QDs are made of a relatively small number of atoms (from 100–10,000 atoms) of semiconductor materials of groups II–VI, III–IV, and IV–VI elements in the periodic table [35]. Their tiny dimension leads to their characterization as “dots”, while “quantum” is due to their properties and behavior being described extensively by quantum mechanics [36].

Quantum dots (QDs) are nanoscale nanomaterials that are said to be zero-dimensional because charge carriers are confined so tightly in three directions [37,38]. Many of their unique properties arise because semiconducting nanocrystals from 2–10 nm diameter are smaller than or equal to their exciton Bohr radius [39,40,41,42,43].

The unique electronic properties of QDs result from the particle size and shape, which can be manipulated for diagnostic purposes. When a QD is excited by an energy photon *hv* (the absorption of light), electrons from the valence band (lower energy level) jump to the conduction band (a higher energy level), resulting in an electron–hole pair called an “exciton”. As they return to the lowest energy state (ground state), electrons and holes recombine and release energy or light in the form of single photons [44,45]. The crystal’s size, composition, and shape determine the wavelength (color) of light that will be released [46]. The larger size QDs emit orange or red wavelengths, while smaller QDs emit shorter blue or green wavelengths. Consequently, the specific tuning of these optical properties (how the QD absorbs and emits energy) can be manipulated to produce distinctive colors by changing the size and shape of the dot [47].

QD semiconducting nanocrystals have an intrinsic band gap, and when light is absorbed, electrons are bridged by excitation. They differ from bulk semiconducting materials due to their inability to create continuous valence and conduction bands, due to the finite number of atoms in a small cluster. Instead, an electronic structure is produced by QDs that is analogous to the discrete electronic states seen in single atoms. Hence, they are also called ‘artificial atoms’ because of their discrete electronic states. As a QD becomes smaller, the band gap becomes larger. That is, there is an increase in the energy level between the higher valence band and the lower conduction band. More energy is further required to excite the dot, and correspondingly, more energy is released when it returns to the ground state [48,49].

QDs are currently studied by many researchers looking to take advantage of their unique optical properties, such as high fluorescence, excellent resistance to photobleaching, small size, and biocompatibility. These properties make them preferable fluorophores compared to conventional organic dyes with broad emission bands that can fade over time [50]. They have generated considerable interest in bioimaging and fluorescence labeling (in vitro and in vivo). Moreover, by adjusting their size and composition, their emission wavelength can be tuned from visible to infrared wavelengths [51,52], which could be useful for in vivo imaging, such as in sentinel lymph node mapping for image-guided surgery.

The surface modification of QDs gives them a potential tool in cancer imaging. The attachment of certain biomolecules (e.g., peptides, antibodies, or small molecules) to QDs can be used in cancer detection and bioimaging [51]. For example, Brunetti et al. created near-infrared (NIR) QDs functionalized with NT4 cancer-selective tetra-branched peptides that were used to produce their specific uptake and selective accumulation at the site of colon cancer [53]. Elsewhere, QDs were reported to aid in revealing in vivo drug release and drug targeting [54,55]. The potential that QDs offer in the fight against cancer is promising.

Inspired by the exceptional features of QDs and the extensive research on their potential and advancement in the field, this review presents basic insights into the properties of QDs and summarizes the different synthesis methods for their production. Then, we discuss the functionalization of QDs, their applications in cancer management, and their cytotoxicity issues, emphasizing the recent research progress mainly in the last 6 years. We guide the reader through the advancements of QDs as a potential cancer imaging and therapy tool with the hope of bridging the gap and leading to novel discoveries in QDs potential in the field of cancer.

## 2. Structural and Optical Properties of QDs

QDs have a structure comprising a core, shell, and sometimes a surface coating, which provides high stability in photo and chemical behaviors, surface activation, and photoluminescence quantum yield. The core is comprised of semiconductor material (e.g., CdSe, CdTe) in a crystal configuration upon which the excitation wavelengths and fluorescence emission are dependent. That core is stabilized by the shell structure that surrounds it. The shell affects the decay kinetics, photostability, and fluorescence quantum yield. A surface layer that can include organic molecules regulates the stability, dispersibility, and potential biological interactions. Initially, when prepared, QDs are generally hydrophobic because they lack surface moieties that form hydrogen bonds; however, hydrophilic molecules or polymers can be attached to confer dispersibility in water. For example, the stability of QDs in water can be increased by the attachment or adsorption of amphiphilic polymers with ionizable functional groups. Figure 1 shows a stylized illustration of a QD.

As mentioned, the structures of typical QDs are core or core/shell structures. Examples of core QD structures include cadmium telluride (CdTe), while core/shell QD structures include CdSe/ZnS or CdTe/CdS, whose properties can be further enhanced via different surface coatings. The electroluminescence and optical properties of the QD core can be manipulated by altering the sizes of the QD core and shell. Furthermore, core/shell QDs having a shell band gap larger than the core band gap give rise to the electroluminescence properties related to exciton decay by radiative processes [56,57].

Quantum Dots exhibit valuable optoelectronic properties due to the quantum confinement effect. These properties include broad absorption spectra, high fluorescence, strong photostability, and size-tunable emission. Larger QDs with large densities of states and band-overlapping structures possess broad absorption spectra and high molar absorptivities. This particular QD property enables efficient excitation of multiple fluorophores using a single light source. Yet, this broad absorption spectrum produces narrow emission spectra due to transitions from a limited number of high energy to low energy levels, which emit very specific photon energies (*hν*). Thus, a light source with a wavelength shorter than the emission wavelength can lead to multiple excitations (and emissions) because of its broad absorption band. These properties that QDs exhibit, broad excitation spectra and narrow emission spectra [57], make them suitable for multiplexed imaging [58,59]. Figure 2 names some optical properties of QDs.

Unlike organic dyes (1–5 ns), the decay rates of the excited state are slower in QDs. For example, after excitation, most QDs exhibit a relatively long fluorescence lifetime–10 to 50 ns–which is advantageous in differentiating QD signals from background fluorescence and attaining more sensitive detection. Thus, time-gated imaging can eliminate background autofluorescence. They also exhibit low photodegradation rates, which is often challenging for organic fluorophores.

Unlike organic fluorophores, which, when exposed to light, bleach after a few seconds of continuous exposure, QDs are quite photostable. Photostability is important in most fluorescence applications. This lack of photobleaching allows continuous or long-term monitoring of slow biological processes [59]. QDs can withstand hours of repeated excitation and fluorescence cycles with high brightness levels and photobleaching thresholds. It has been observed that QDs are more photostable than “stable” organic dyes such as Alexa488 [60], and thus offer several advantages in diagnostic applications [61,62].

As mentioned, the size- and chemically-tunable properties are advantageous in selecting an emission wavelength suitable to a specific experiment. For example, the emission wavelength of cadmium sulfide (CdS) and zinc selenide (ZnSe) dots can be tuned from blue to near-ultraviolet light. Similarly, cadmium selenide (CdSe) QDs of different sizes emit light across the visible spectrum. For far-infrared and near-infrared emissions, indium phosphide (InP) and indium arsenide (InAs) QDs can be used [63]. Table 1 lists the emission ranges for some common QDs.

These unique optical properties of QDs make them highly appealing to a wide array of research and diagnostic applications in diagnostic bioimaging, drug delivery, and more.

## 3. Synthesis of QDs

QDs must be carefully synthesized to meet specific optical requirements. Their synthesis can be divided into two general categories, the top-down method and the bottom-up approach [56].

### 3.1. Top-Down Approach

In the top-down approach, QDs are formed by the ablation of bulk semiconductor materials. This includes processes such as electron beam lithography, reactive ion etching, and focused ion beam. These processes synthesize QDs with diameters of around 30 nm. However, these processes have limitations, such as incorporating impurities during synthesis.

#### 3.1.1. Electron Beam Lithography (EBL)

In electron beam lithography (EBL), the surface of a resist (electron-sensitive material) is patterned by scanning with a focused beam of electrons. The resist is made of a polymeric compound, which can either be a negative resist (i.e., long-chain polymer) or a positive resist (i.e., short-chain polymer). The solubility of the resist is altered by the electron beam, allowing the selective removal of either exposed regions or non-exposed regions of the resist when immersed in a solvent (called a developer). If the resist becomes soluble when immersed, it is a positive resist; if it becomes insoluble (i.e., unexposed parts removed), it is a negative resist. The purpose is to fabricate very small structures in the resist whose pattern can then be transferred to the substrate by etching. Although this technique can design patterns directly with sub-10 nm resolution, it is slow and expensive [65,66]. Nandwana et al. [67] reported direct patterning of QD nanostructures using EBL. In this example, functionalized CdSe/ZnS QDs were deposited onto a gold-coated silicon substrate, followed by direct patterning using EBL in the QD film. The QD film was washed using toluene, which removed the unexposed QDs, leaving the exposed areas anchored to the substrate due to the electron beam. QDs were observed to retain their optical properties after cross-linking. Similarly, Palankar et al. [68] reported using EBL to generate QD micropattern arrays. The QDs fabricated were reported to retain their fluorescence and bio-affinity during lithography.

#### 3.1.2. Reactive Ion Etching

In dry etching, an etching chamber is used, where a reactive gas species is introduced, and plasma is formed by applying radio frequency energy by which the gas molecules are broken into reactive fragments. These high-energy species collide with the surface, reacting to form a volatile reaction product. Thus, the surface is slowly etched away. The surface can be protected from etching with a mask pattern. This process is also referred to as reactive ion etching [66,69,70]. Site- and dimension-controlled indium gallium nitride (InGaN) QDs were fabricated by Lee et al. [71]. The QDs were disk-shaped and integrated into a nanoscale pillar. They utilized inductively coupled plasma reactive ion etching to fabricate arrays of nanopillars with different densities and nanopillar diameters from InGaN/GaN. They observed single nanopillars that exhibited strong and distinct photoluminescence at room temperature. The advantages of this process include reducing the amount of etchants used, easy disposal, and eliminating the need to use dangerous liquid etchants. However, the drawback of this process is that it is both time-consuming and expensive, as it requires very specialized equipment [66].

#### 3.1.3. Focused Ion Beam (FIB)

QDs can be fabricated with exceedingly high lateral precision through the focused ion beam technique. The semiconductor substrate’s surface is sputtered using highly focused beams from a source of metal ions (Au/Si, Ga). The size, shape, and inter-particle distances of the QDs depend on the ion beam size. Furthermore, it has been reported that a beam with a minimum diameter of 8–20 nm allows QDs to be etched to <100 nm [70]. Choi et al. [72] used focused ion beam luminescence quenching (FIB-LQ) to enhance the single photon purity of the site-controlled QD emission. Optical quality was retained while the SNR of the QD improved, and at increased temperatures, single photon properties were maintained due to the improved signal-to-noise ratio (SNR). In a similar study, Zhang et al. [73] combined focused ion beam (FIB) patterning and self-assembly quantum dots to produce regular QD arrays. High resolution and high flexibility are the advantages of this process. However, the technique is slow and utilizes expensive equipment.

### 3.2. Bottom-Up Approach

In the bottom-up approach, small units are assembled (precipitated) into the desired structure’s shape and size. This process involves nucleation, growth, and chemical decomposition [74]. QDs are synthesized with different techniques, which are further classified into wet-chemical and vapor-phase methods. Wet-chemical methods processes include sol–gel, and microemulsion, while vapor-phase methods processes include molecular beam epitaxy, physical vapor deposition, and sputtering [70]. In wet-chemical methods, conventional precipitation methods are followed by measured control of single solution parameters or a mixture of solutions. The process of precipitation always involves both nanoparticle nucleation and limited growth. Nucleation can involve homogenous, heterogenous, or secondary nucleation. QDs of the desired size, shape, and composition can be acquired by varying factors such as stabilizers, temperature, electrostatic double-layer thickness, and precursor concentration [70,75]. More details are given below.

#### 3.2.1. Wet Chemical Methods

##### Sol–Gel

Sol–gel methods are commonly used to synthesize QDs [76,77]. The technique prepares a sol (a solution or suspension) of a metal precursor salt (acetates or nitrates, alkoxides) in a base or acidic medium. The process has three steps: hydrolysis, condensation (formation of sol), and growth (formation of gel). In brief, inside the solvent medium, the metal precursor hydrolyzes and condenses, thereby forming a sol, which then grows or polymerizes, forming a network (gel). This process can be used to prepare thin films, fibers, microspheres, etc. The advantages of this process for QD formation include good control of composition, better control of structure, incorporation of nanosized materials, and no use of special or expensive equipment. However, the process is slow, complex, and may involve toxic solvents [78]. QDs of semiconductor types II–VI and IV–VI zinc oxide, cadmium sulfide, and lead sulfide (ZnO, CdS, PbS) have been synthesized using this method [76,77,79]. For example, mixing solutions of Zn-acetate with alcohol and sodium hydroxide, followed by controlled aging in air, produced zinc oxide (ZnO) QDs [76]. Titanium dioxide (TiO_2_) QDs were synthesized by Javed et al. [80] using the sol–gel reflux condensation method. They reported the QD to have an average 5–7 nm crystallite size, which offers a large surface area and exhibits photocatalytic properties. In another study by Jiang et al. [81], zinc selenide (ZnSe) QDs embedded in silicon oxide (SiO_2_) thin films were synthesized using the sol–gel process. The synthesis was done with H_2_SeO_4_ as a source for selenium and Zn (Ac)_2_·H_2_O as a source for zinc. One advantage of this approach to making ZnSe/SiO_2_ thin films is a reduction in the amount of selenium volatilization. The sol–gel process was reported to be cost-effective and simple [80,81,82].

##### Microemulsion Process

A useful method for synthesizing QDs at room temperature is the microemulsion process. Two microemulsions of an aqueous phase in oil are prepared, each having a single chemical component of the semiconductor. While mixing slowly at room temperature, the water droplets collide and merge, thereby creating a mixture that forms QDs inside the very small water droplet. The process can also be done using an oil-in-water emulsion with the oil phase containing the semiconductor components. The use of alcohol instead of water has also been employed. In the reverse microemulsion process, water is dispersed into oil (immiscible liquid) and stirred vigorously in the presence of a surfactant to form extremely small emulsion droplets. The variation of the water-to-surfactant molar ratio controls the size of the water droplet, which in turn affects the size of the resulting QD [44,70,83]. The reverse micelle method has been used to prepare II–VI core and core/shell QDs. Shakur [84] synthesized zinc sulfide (ZnS) QDs by the reverse micelle method using polyvinyl pyrrolidone as a surfactant and produced a size of 2.1 nm. In another study, Karanikolos et al. [85] synthesized luminescent zinc selenide (ZnSe) QDs using a microemulsion process. The synthesized QDs were reported to exhibit excellent photostability and size-dependent luminescence. Cadmium sulfide (CdS) and CdS/ZnS semiconductor QDs were synthesized by the reverse micelle method in a study by Lien et al. [86]. Sodium bis (2-ethylhexyl) sulfosuccinate (AOT) was used as a surfactant. The synthesized QD had a diameter of ~2.5 to 4 nm, which was dependent on the surfactant concentration. In addition, the core/shell nanocrystal structure was reported to have excellent luminescence and photostability. This process is said to be cost-effective, easy to handle/control by modifying parameters such as the ratio of water to surfactant, inexpensive, highly reproducible, and displays good monodispersity [87,88,89]. However, this process has limitations, such as low yield and the need for large amounts of surfactant, which could result in the incorporation of impurities and presents difficulty in separating the surfactant from the final QDs [90].

#### 3.2.2. Vapor-Phase Method

Vapor-phase methods to produce QDs involve QDs deposited in an atom-by-atom process, as described below.

##### Molecular Beam Epitaxy (MBE)

Molecular beam epitaxy (MBE) is one of the vapor-phase methods used under ultra-high vacuum conditions (~10^−10^ Torr). It involves the deposition of overlayers to grow elemental compound semiconductor materials of nanostructures on a heated substrate [91]. The process forms a beam of atoms or molecules from the evaporation of an apertured source. The beams can be formed from solids (Ga and As to form GaAs QDs) or a combination of solids and gases (e.g., PH_3_ or tri-ethyl gallium). This method uses the large lattice mismatch to self-assemble QDs from II–VI semiconductors and III–V semiconductors [70]. During the process, a reflection high-energy electron diffraction gun is used to monitor the growth of the crystals. Although it is expensive and requires complex equipment, heating the material is slow and controlled, and the process does not involve a slow chemical reaction, resulting in a reduced amount of defects [66]. Brault et al. [92] used molecular beam epitaxy to grow Al*_y_*Ga_1−*y*_N QDs on Al*_x_*Ga_1−*x*_N (0001) for light-emitting-diode applications.

##### Physical Vapor Deposition (PVD)

Physical vapor deposition requires a high vacuum (≤10^−6^ Torr) to retain a good vapor flow. A material is sublimated inside the vacuum by thermal evaporation, thereby condensing the substrate from the vapor. Techniques such as resistive heating, electron beam heating, and laser ablation have been used to evaporate the material. The quality of the films produced and their physical characteristics are influenced by the rate of deposition, pressure, substrate temperature, and distance between source and substrate. These factors control the creation of QDs from the thin films deposited [70]. As an example, niobium pentoxide (Nb_2_O_5_) QDs were grown by Dhawan et al. using PVD [93]. This process does not require expensive chemical reagents, the coatings by PVD have excellent adhesion, and the process allows the deposition of different types of materials. However, the equipment employed is complex and expensive [94,95].

##### Sputtering

The sputtering process produces nanostructures by bombarding a surface with high-energy particles (e.g., via gas or plasma). It is an effective technique for developing thin films of nanomaterials. During the process, high-energy gaseous ions bombard the semiconductor surface (target material), causing the physical expulsion of atoms or molecules from the surface, depending on the incident gaseous ion energy [96,97]. This technique is also referred to as ion sputtering and is commonly performed in an evacuated chamber. The process is done in different ways, such as radio frequency and magnetron sputtering [94,98,99]. Cadmium selenide (CdSe) QDs were synthesized by Dahi et al. [100] using radio frequency magnetron sputtering. The synthesized QDs had an average size of less than 10 nm in diameter using a radio frequency power of 14 W and a deposition duration of 7.5 min. It is noteworthy that increasing either RF power or deposition time (or both) increased the CdSe QD size. The advantages of this process are reduced surface contamination, no required solvents, and facile tuning of the size, shape, and optical density through careful control of pressure, temperature, and deposition time. However, a drawback of this process is the redeposition of unwanted atoms, which may contaminate the QDs.

### 3.3. Other Syntheses

QDs are also produced using hydrothermal synthesis. This is a one-pot synthesis by which inorganic salts are crystallized from aqueous solution by regulating temperature and pressure. In this technique, the temperature can be raised very high due to the pressure containment in the autoclave. This results in partial chemical decomposition and promotes molecular collisions, causing the formation of QD. By changing the pressure, temperature, reactants, and aging time, different QD sizes and shapes can be attained [70]. This method of preparing QDs gives excellent photostability and high quantum yield. The process is efficient, timesaving, and more convenient. However, a significant disadvantage of this process is the need for expensive autoclaves [101]. Shen et al. [102] developed nitrogen-doped carbon QDs (N-CQDs) by the hydrothermal synthesis of glucose and phenylenediamine. The synthesized N-CQDS were reported to have good photostability, water solubility, and low toxicity. They were also reported to be excellent fluorescent probes for Fe^3+^ and CrO_4_^2−^ in addition to serving as cell imaging reagents for Hela cells. Likewise, QDs can be fabricated using the solvothermal method, which is similar to the hydrothermal except that organic solvents with high boiling points are used instead of water [103,104]. Luo et al. [105] synthesized multiple color emission iron disulfide (FeS_2_) QDs by the solvothermal method. Temperature, time, and the reactant ratio were varied to make QDs with blue, green, yellow, and red fluorescence. The blue emission of the QDs was used as a fluorescent responsive signal and the yellow emission was used as a reference signal to construct a molecular imprinting radiometric sensor used for the visual detection of aconitine. The process was reported to be simple and low in cost.

The microwave-assisted synthesis of QDs is a rapid heating method that shortens reaction time and improves production yield. In this method, fewer solvents are used, and tiny particles with a narrow size distribution can be created [106,107]. Cadmium selenide (CdSe) QDs were synthesized by Abolghasemi et al. [108] using the microwave-assisted method. The QDs were synthesized in an N-methyl-2-pyrrolidone solvent with a microwave irradiation power of 900 W. It was reported that this method showed easy control of the size and band gap energy of the QDs, resulting in controllable emission from photoluminescence spectroscopy. The performance of the QDs was tested in photovoltaic solar cells, where results showed that the QDs are suitable sensitizers.

Recently, an ultrasonic technique was employed to synthesize QDs. This method utilizes ultrasound, which causes acoustic cavitation. This involves the formation, development, and implosive collapse of bubbles in a liquid, which produces high pressure and high energy [109,110]. Graphene QDs (GQDs) were synthesized by Zhu et al. [111] from graphene oxide (GO) by ultrasonication in KMnO_4_ for 4 h. High-resolution transmission electron microscopy (HR-TEM) revealed that the GQDs had an average of 3.0 nm lateral diameter with a narrow size distribution. The GQDs were reported to be uniform and of high crystallinity. These QDs were used in an alkaline phosphate (ALP) activity assay. In another study by Chen et al. [112], perovskite QDs were synthesized using ultrasonic synthesis. This synthesis method was reported to produce smaller particle sizes with a more uniform particle-size distribution. They also used this method to prepare different chemical compositions of CH_3_NH_3_PbX_3_ QDs that could tune emission wavelengths, thus providing a wider range of pure colors. Table 2 catalogs the various types of QDs and their synthesis.

## 4. Surface Functionalization of QDs

QDs have been widely used in various applications such as bioimaging, drug delivery, and diagnostics [125,126,127,128,129,130]. This has only been possible due to functionalizing their surfaces, thereby enhancing biocompatibility, uptake, stability, and reducing biological toxicity [131]. After synthesis, QDs are generally hydrophobic, which could produce a cytotoxic effect on cells or reduce their uptake efficiency, limiting their use in clinical practice. Hence, the surfaces of QDs need to be altered for prospective diagnostic and therapeutic applications by making them hydrophilic, and by attaching various chemical groups and targeting molecules [132,133,134]. This can be achieved by coating or conjugating the surface of the QDs with molecular ligands, growing silica, or applying other coatings to the QDs, such as with amphiphilic polymers [135,136,137,138]. The next sections present general descriptions of methods for surface modification.

### 4.1. Ligand Exchange

This process involves exchanging hydrophobic ligands such as trioctylphosphine oxide (TOPO), trioctylphosphine (TOP), and hexadecyl amine (HDA) on the QD surface with hydrophilic ligands to promote the formation of stable suspensions in water [139]. The most common approach for ligand exchange is the use of thiols (-SH), such as mercaptoacetic acid (MAA), mercaptopropionic acid (MPA), mercaptoundecanoic acid (MUA), and dihydrolipoic acid (DHLA) as anchoring groups, all of which present carboxyl (-COOH) groups as hydrophilic and ionized groups to enhance hydrogen bonding with water. Furthermore, at the proper pH (pH 5 to 12), ionic groups provide charge repulsion between particles. The attachment of hydrophilic polymers such as PEG can enhance the solubility range of QDs by steric repulsion [51,139,140].

The as-synthesized QDs are reported to have a small hydrodynamic size, which is useful in fluorescence resonance energy transfer (FRET) experiments [141]. However, after the process, there is a decrease in fluorescence quantum yield.

In other studies, the multidentate ligands were used as sensing probes to detect bovine serum albumin (BSA) protein in aqueous media [142]. Similarly, Chen et al. [143] reported the ligand exchange of oleate-capped ZB-CdSe with oleylamine, resulting in a significant decrease in photoluminescence quantum yield (PLQY). In another study [144], a method was optimized to overcome the issue of the reduced fluorescence and stability of silver telluride (Ag_2_Te) QDs. Tributylphosphine (TBP) was added during synthesis, which was used as a precursor (TBP-Te) to form a high fluorescent Ag_2_Te core. The rapid injection of TBP-Te precursor in hot solvent resulted in a PLQY of up to 6.51%. This was then followed by phase transfer of NIR-II Ag_2_Te QDs via direct ligand exchange of hydrophobic Ag_2_Te surface ligands with ligands of the thiol family (e.g., glutathione (GSH), DL-cystine, dithiothreitol (DTT), dihydrolipoic acid (DHLA), DHLA-EA, cysteamine, and thiol-containing PEG). It was observed that the hydrophilic thiol ligands promoted the water solubility of QDs and that only ligands composed of free thiol groups were suitable for this technique. Moreover, the QDs were reported to retain a PLQY of nearly 5% as well as exhibiting good biocompatibility. PEGylated Ag_2_Te QDs were used for “second” near-infrared (NIR-II) imaging in mice. Unlike near-infrared (NIR) imaging with emission wavelengths between 700–900 nm, which is reported to produce substantial background signal and affect the quality of images [145], the NIR-II window encompasses emission wavelengths between 1000–1700 nm. Thus, it exhibits excellent penetration capacity and high-resolution fluorescence imaging in the living body. Real-time imaging in mice showed high brightness in abdominal vessels, sacral lymph nodes, hindlimb arterial vessels, and tumor vessels [144].

### 4.2. Surface Silanization

This coating process produces a silica shell around the QDs. It is an effective process for the modification of hydroxyl-rich material surfaces. This technique initially deposits hydroxyl groups by ligand exchange of the surface hydrophobic groups with a thiol-derived silane ligand (e.g., mercaptopropyltris (methyloxy)silane (MPS)) to place silanol groups on the surface. This is followed by further silica shell growth, where other silanes can be added on the outer surface to modify the surface charge or provide reactive functional sites. Aminopropylsilanes (APS), phosphosilanes, and polyethylene glycol (PEG)-silane are the most frequently used silanes [138,140]. Due to the silica thickness, the aqueous stability, size, biocompatibility, and fluorescence of the QDs are enhanced after being covered with a silica layer [146]. The layer also serves as a platform for further coating processes due to the silane shell end terminal groups exposing either their thiol, phosphate, or methyl terminal ends for subsequent reactions [147]. The advantage of this process is that the silica shells are highly crosslinked, thereby stabilizing the silanized QDs [147]. Furthermore, this is a preferred approach because the QDs can be made more biocompatible, less toxic, and chemically inert. The presence of silica increases the photostability of QDs by preventing surface oxidation [148,149], which makes them useful for applications such as drug and gene delivery, therapy, and bioimaging [150,151].

For example, silica coating is reported to suppress photoluminescence bleaching through the reduction in photochemical oxidation of cadmium selenide (CdSe) surfaces [152]. Similarly, encapsulation in silica was reported to prevent the loss of Cd^2+^ ions [153]. However, the silica shell is reported to increase the hydrodynamic size. Ham et al. [154] fabricated SiO_2_@InP QDs@SiO_2_ NPs by encapsulating multiple indium phosphide/zinc sulfide (InP/ZnS) QDs onto silica templates and coating silica shells over them. The fabricated QDs were reported to exhibit hydrophilic properties due to the surface silica shell. The NPs were further applied in detecting tumors where the fluorescence signal was notably detected in the tumor. Goftman et al. synthesized silica-coated cadmium selenide (CdSe) QDs by the reverse microemulsion method. The silica-capped QDs were reported to have high stability and initial brightness [155].

### 4.3. Amphiphilic Ligands

In this approach, the hydrophobic surfactants trioctylphosphine, trioctylphosphine oxide, and hexadecylamine (TOP/TOPO/HDA) are preserved on the surface of the QDs. They are coated or encapsulated with crosslinked amphiphilic polymers containing hydrophobic and hydrophilic segments. The synthesized QDs are hydrophobic, and upon encapsulation with an amphiphilic polymer, an attraction is formed between the hydrophobic alkyl chains and the hydrophobic components of surfactants on the surface of the QDs. In contrast, the hydrophilic component (carboxylic acid or polyethylene glycol chains) provides dispersibility in aqueous solution and chemical functionality. During the coating process, the amphiphilic polymers are hypothesized to provide additional stability to the QDs through crosslinking reactions [51,138,156]. Some amphiphilic polymers include poly (acrylic acid), phospholipids, and maleic anhydride copolymers [156].

Yoon et al. [157] fabricated CdSe@ZnS/ZnS core/shell QDs encapsulated with an amphiphilic polymer (i.e., poly(styrene-co-maleic anhydride) PSMA). The amphiphilic polymer (PSMA) served as a crosslinker for the matrix polymer between the maleic anhydride of QDs and the diamines of PDMS within a ring-opening reaction. This produced a highly transparent polymer at low curing temperature with enhanced compatibility between QDs and a polydimethylsiloxane (PDMS) matrix and also improved dispersion of QDs. The encapsulated QDs were also reported to preserve photoluminescence intensity as a result of using this encapsulation method. They further fabricated a light-emitting diode, which was observed to have excellent luminous efficacy.

Starch-g-poly(acrylic acid)/ZnSe-QDs hydrogel was fabricated by Abdolahi et al. [158]. The QDs were fabricated to serve as an effective adsorbent and photocatalyst. In another study, Speranskaya et al. [159] synthesized hydrophobic cadmium selenide (CdSe)-based QDs. The QDs were hydrophilized by coating with amphiphilic polymers (i.e., maleic anhydride-based polymers and Jeffamines). The polymer-coated QDs were reported to retain up to 90% of their initial brightness. Carolina and Wolfgang [160] synthesized pyridyl-modified amphiphilic polymeric ligands (Py-PMA) in order to overcome the limitations of QDs coated with amphiphilic polymers, such as a decrease in photoluminescence quantum yield and diffusion of small molecules causing oxidation. Poly (isobutylene-alt-maleic anhydride) backbone was used for synthesis with pyridyl and alkyl end groups. The synthesized polymer-coated QDs were reported to preserve photoluminescence quantum yield and exhibit good colloidal stability in water.

### 4.4. Microsphere Coating

The microsphere coating of QDs is of great interest in biological applications [161]. The formation of composite nanostructures in which micro-composite nanostructures are assembled from QDs by an encapsulant component that can serve as a glue, a scaffold, or a matrix has been developed by researchers. Different techniques have been used for encapsulating QDs into microspheres. These include dispersing synthesized microspheres, placing QDs in a solvent or non-solvent mixture [162], and electrostatic bondage of QD to the microsphere surface [163]. The reverse microemulsion method [164] and emulsion polymerization [165] are encapsulation techniques. For example, a uniform magnetic/fluorescent microsphere was synthesized by Li et al. [164] using the Pickering emulsion polymerization method. The authors synthesized QD-encoded magnetic microbeads that were closely covered with a Pickering structure containing many silica nanoparticles. This was done using a microfluidic device that produced homogenous microbeads by forming Pickering emulsion droplets. The oil-in-water emulsion (O/W) droplets fabricated contained the oil phase (i.e., Fe_3_O_4_ NPs and QDs along with PSMA polymer were dispersed in toluene) and the water phase (silica NPs dispersed in deionized water), with the silica NPs accumulated at the interface (i.e., the oil and water interface). Thus, the silica NPs served as stabilizers. They reported the successful synthesis of a CdSe/ZnS core/shell along with a Fe_3_O_4_ nanoparticle encapsulated in a magnetic fluorescence microsphere (MFM microsphere). The microspheres were observed to be highly homogenous in shape, to have a high surface area, and to be well dispersed. Moreover, they also exhibited excellent fluorescent stability under room temperature. Hence, they further tested the microspheres to detect tumor markers (CEA, CA199, CA125) in a single sample. Results showed the detection limits achieved to be 0.027 ng/mL, 1.09 KU/L, and 1.48 KU/L for CEA, CA199, and CA125, respectively. The microspheres exhibited excellent detection performance.

Zhao et al. [166] synthesized bismuth oxybromide/carbon quantum dots (BiOBr/CQDs) microspheres using the solvothermal method followed by the hydrothermal process. The synthesized microsphere QDs were reported to exhibit excellent photoactivity under visible light irradiation due to exceptional electron transfer, and the CQDs exhibited increased light harvesting capacity in addition to stability and enhanced visible-light absorption ability. Moreover, QD-based sensors have been observed to agglomerate, leading to self-absorption and non-radiative deactivation. Hence, to overcome this issue, microsphere-QD-sensor platforms are being utilized. For instance, Khan et al. [167] developed a fluorescent sensor platform for heavy metal sensing. The authors used non-toxic fluorescent zinc oxide ZnO–QDs that were conjugated with carboxymethyl cellulose (CMC) polymer (ZCM) for the synthesis of microspheres for sensing heavy metal (cationic metal ions, e.g., Pb^2+^, Hg^2+^, Fe^3^+, Cr^6+^, Cu^2+^, Ni^2+^, Mn^2+^). To differentiate these metal ions, a fluorescence turn-off response was adopted. Their results showed that the developed sensor had an affinity towards the different heavy metal ions and excellent photostability. In addition to detecting the heavy metals, the sensor could also quantify them with an accuracy of 5%. However, only Fe^3+^, Cr^6+^, and Cu^2+^, among the seven metals, showed high sensitivity toward the sensor system. Table 3 presents several examples of functionalization of the surface of QDs.

This table shows that each of these methods has its advantages. However, the final choice depends on the specific application and the requirements. For instance, the ligand exchange process decreased the photoluminescence quantum yield (PLQY). Hence, direct encapsulation of QDs with silica shell resolves the issue of reduced luminescence yields. This layer of silica on the QD is reported to provide enhanced aqueous stability and fluorescence by the silica’s thickness [146]. Yet, it was reported that coating with silica shell yields larger QDs due to the difficulty in controlling the silica thickness [173]. Moreover, the encapsulation of QDs with an amphiphilic polymer also preserves quantum yield (QY) even after surface modification.

Regarding microspheres, they are reported to provide hydrophobic protection. This is because some QDs are hydrophobic in nature and not biologically useful. Thus, QDs are functionalized or coated to make them water-dispersible and enhance their biocompatibility. However, it was reported that the size of the photoluminescence (PL) microsphere determined QD stability, with a larger PL microsphere observed to give more hydrophobic protection of the interiors of QDs compared to smaller PL microspheres [178]. In other words, for every possible application, the prerequisite is to properly functionalize the surface of the QDs accordingly while ensuring they do not lose their physicochemical properties, which are enhanced in aqueous media. Figure 3 illustrates some surface functionalization approaches of a multifunctional QD.

## 5. Application of QDs

### 5.1. QDs for In Vitro Tumor Imaging

One of the most important applications of QDs in recent research has been to produce in vitro fluorescent images of cancerous cells. The unique properties of QDs make them preferable to traditional fluorescence organic dyes. A schematic representation of QDs for in vitro tumor imaging is shown in Figure 4.

Nitrogen-doped carbon QDs (N-CQDs) were synthesized hydrothermally by Wu et al. [179] using tetraphenyl porphyrin and its metal complex (Pd or Pt) as a precursor. As a result of the strong photoluminescence (PL) exhibited by the CQDs, they were investigated as imaging probes for living cells. HeLa cells treated with CQDs (0.2 mg/mL) exhibited blue, green, and red fluorescence at excitation wavelengths of 405 nm, 458 nm, and 514 nm. Fluorescence images showed CQDs to be mainly dispersed in the cell cytoplasm, and the nucleus showed weak emission signals. These experiments supported that CQDs enter into cells via endocytosis.

Near-infrared (NIR) emitting CdHgTe/CdS/CdZnS QDs were synthesized by Liu et al. [180]. The QDs were coated with N-acetyl-L-cysteine (NAC), 3-mercaptopropionic acid (MPA), and thioglycolic acid (TGA) thiol ligands. HeLa cells were stained with these QDs and exposed to continuous UV excitation. In vitro studies showed that after 20 min of irradiation, stained HeLa cells produced red emission. Fluorescence images revealed that after 40 min, NAC-tagged CdHgTe/CdS/CdZnS QD-stained cells showed high photostability in the intracellular environment compared to TGA- and MPA-capped QDs. This success was attributed to the NAC thiol capping of the QDs preventing degradation.

Near-infrared (NIR) CdTe/CdS was synthesized in an aqueous solution with 3-mercaptopropionic acid (MPA) as a stabilizer. These QDs were employed to monitor the change in Cu^2+^ concentration in living cells. HeLa cells were incubated with the synthesized QDs (5 µg/mL), followed by adding Cu^2+^ (30 µM) before fluorescence imaging. A bright fluorescence signal from the cells at 700–800 nm showed efficient uptake of CdTe/CdS. However, when HeLa cells were treated with 30 µM of Cu^2+^ before incubation with the QDs, significant fluorescence quenching (~90%) was observed. This observation was attributed to the aggregation of QDs mediated by the competitive binding between MPA and the Cu^2+^ in the solution. Overall, they reported the nanosensor to exhibit high selectivity, excellent photostability, and rapid response [181]. Fluorescence images generated during this study are shown in Figure 5.

Shi et al. synthesized molybdenum disulfide (MoS_2_) QDs with Na_2_MoO_4_ as the molybdenum source and 2H_2_O·GSH as the sulfur source using hydrothermal synthesis [182]. The reaction conditions (i.e., precursor, precursor ratio, ratio, reaction time, and temperature) were optimized to improve the photoluminescence quantum yield (PLQY). These MoS_2_ QDs were then used for fluorescence imaging. The in vitro studies reported glutathione–molybdenum disulfide (GSH-MoS_2_) to be biocompatible after SW480 cells were exposed to the QDs (from 0 to 1.5 µM Mo). They reported that blue fluorescence was observed in the SW480 cells cytoplasm.

In another study, blue-fluorescent nitrogen-doped graphene quantum dots (N-GQDs) were produced by Tao et al. [183]. The QDs were synthesized using hydrothermal synthesis from citric acid and diethylamine, and the binding sites were highlighted. The doping with nitrogen element resulted in ample amide II bonds (this provides a structure for integrating HA with N-GQDs) and enough binding sites to conjugate hyaluronic acid (HA). In order to recognize the breast cancer cells (MCF-7 cells), the N-GQDs were conjugated to HA through an amide bond. It was reported that the formation of amide bonds was more conducive under alkaline conditions. In addition, MCF-7 cells exhibited stronger fluorescence as a result of combining HA-conjugated N-GQDs (HA-N-GQDs) with CD44 over-expressed on the MCF-7 cells surface. Their results showed the good cytocompatibility, low toxicity, and high fluorescence of HA-N-GQDs.

### 5.2. QDs for In Vivo Tumor Imaging

The excellent fluorescent signals and multiplex capabilities of QDs make them a promising tool for cancer bioimaging, specifically in vivo. Researchers have reported many examples of using QDs to image tumors in vivo. A schematic representation is shown in Figure 6.

For instance, Zhu et al. [184] developed near-infrared (NIR) fluorescent silver selenide (Ag_2_Se) QDs tagged with Cetuximab for targeted imaging and cancer therapy. The multifunctional nanoprobe was reported to display fluorescent contrast at the tumor site, and 24 h post-injection, the fluorescence was still easily detected at the tumor site, unlike with Ag_2_Se QDs alone. Their results showed that this nanoprobe significantly inhibited tumor growth, and the survival rate of nude mice with orthotopic tongue cancer improved from 0% to 57.1%. This platform was claimed to have successfully targeted orthotopic tongue cancer.

Sulfonic-graphene QDs were used by Yao et al. [185] to target tumor cells in vivo. They showed that the sulfonic-GQDs had successfully penetrated the plasma membrane into tumor cells without modifying any bio-ligand, which they attributed to high interstitial fluid pressure. They also reported fluorescence of the sulfonic-GQDs at an excitation of 470 nm in tumor-bearing mice post-injection. Rapid accumulation of sulfonic-GQDs at the tumor site occurred 0.5 h after injection and was cleared 24 h later. This research demonstrated sulfonic-GQDs’ ability to target nuclei of tumor cells in vivo with a low distribution in normal tissues.

In another study, Wu et al. [186] developed a novel strategy against tumor cells. They modified near-infrared fluorescent indium phosphide (InP) QDs using a vascular endothelial growth factor receptor 2 (anti-VEGFR_2_) monoclonal antibody and attached miR-92a inhibitor to VEGFR_2_-InP QDs. The miR-92a is said to enhance the expression of tumor suppressor p63. Their results showed that the functionalized InP nanocomposite showed an enhanced NIR fluorescence intensity at the tumor site, which had accumulated via enhanced permeability and retention effect, thereby targeting tumor angiogenic cells. Moreover, using nude mice inoculated with k562 cells, they investigated the suppression of tumor growth in vivo. They observed the functionalized InP nanocomposite to significantly inhibit tumor growth compared to InP QDs or miR-92a, which showed moderate suppression. Overall, the developed system may provide a new and promising chemotherapy strategy against tumor cells.

Fluorescent silver indium sulfide/zinc sulfide (Ag-In-S/ZnS (AIS/ZnS)) QDs with red emission were synthesized by Sun et al. [187] and then dispersed with poly(vinylpyrrolidone) (PVP) for imaging of tumor drainage lymph nodes. The synthesized QDs were subcutaneously injected in nude mice, and a bright red fluorescence was observed, suggesting that AIS/ZnS QDs are excellent fluorescent probes for in vivo imaging. To image sentinel lymph nodes, AIS/ZnS QDs were intradermally injected into the extremities of nude mice, and the QDs were observed to migrate to sentinel lymph nodes. Furthermore, within 10 min of intratumoral injection in mice bearing H460 tumors, AIS/ZnS QDs were observed to stain tumor drainage lymph nodes with bright red fluorescence. However, after 10 min, only weak fluorescence was observed in the tumor drainage lymph node.

Triple-negative breast cancer (TNBC) is known to develop rapidly and is associated with recurrence and metastasis. The efficacy of chemotherapy is reported to be poor, with the survival rate of patients affected being less than 30%. Hence, Zhao et al. [188] designed and constructed biomimetic black phosphorus QDs (BBPQDs) coated with cancer cell membranes for tumor-targeted photothermal therapy (PTT) and anti-PD-L1 mediated immunotherapy. The stability of the BBPQDs after encapsulating with cancer cell membrane exhibited active targeting and enrichment ability in tumors. Subsequently, Cy5.5-labelled BBPQDs were intravenously injected into BALB/c mice bearing 4T1 tumors to investigate tumor targeting and tissue distribution. The BBPQDs were reported to exhibit significant fluorescence intensity post-injection compared to Cy5.5-labeled BPQDs. Moreover, after 72 h, the BBPQDs showed good tumor targeting, high aggregation, and good retention at the tumor site. The BBPQDs exhibited excellent photothermal properties and could kill tumors directly and induce dendritic cell maturation and the activation of T cells. BBPQD-mediated PTT and αPD-L1 combined inhibited tumor recurrence and metastasis through the immune memory effect.

Stable fluorescent CQDs were synthesized by Huang et al. [189] under photobleaching treatment. The synthesized CQDs were reported to have a quantum yield (QY) of ~13% at an excitation of 365 nm, proving them to be viable in bioimaging mice with Smmc-7721 tumor cells. The CQDs were intravenously injected (0.2 µg/mL). Optical images of the distribution of the CQDs were obtained at different time points. The study reported detecting fluorescence signal 5 min post-injection, and CQDs accumulated at the tumor site after 3 h. Complete accumulation of the CQDs was reported to occur at 12 h. The CQDs appeared to exhibit good biocompatibility and could be used for a prolonged imaging period. Results also showed that CQDs accumulated in the tumor, kidney, and liver. However, no fluorescence signal was detected in the heart, lungs, and spleen. In addition, the CQDs were reported to exhibit excellent bioimaging performance, low cytotoxicity, and antioxidant activity.

Although the unique optical properties of QDs make them an attractive fluorescent probe, specifically in bioimaging, the potential toxicity of QDs, such as those containing toxic heavy metals, has limited their applications. Hence, Yaghini et al. [190] developed a heavy-metal-free biocompatible and good photoluminescence quantum yield (PLQY) Indium-based QD (bio CFQD^®^ NP) for imaging in vivo. These metal-free QDs were investigated for in vivo axillary lymphatic mapping applications. Twenty-four hours post-injection of the QDs in the paw of rats, the QDs were observed to accumulate mainly in the regional lymph nodes with negligible accumulation in the spleen and liver while exhibiting stable photoluminescence. Their low intrinsic toxicity makes them attractive for in vivo tumor imaging.

### 5.3. QDs for Drug Delivery

QDs are just one example of the numerous nanoparticles (NPs) that have been widely investigated for drug delivery applications. Reports show that antitumor efficacy is increased while systemic side effect is reduced, which is attributed to effective nanoparticle entrapment of anti-cancer drugs and control of distribution in cells and in tissue. The use of nanoparticles as drug delivery agents has been reported to overcome the limitations posed by traditional cancer therapies, including but not limited to overcoming multidrug resistance, lack of specificity, and cytotoxicity. Their specific advantages, such as enhanced stability, reduced toxicity, precise targeting, and biocompatibility, promote the use of NPs as nanocarriers in cancer therapy [191,192,193].

Moreover, these nanocarriers have been found to facilitate the administrative routes and enhance the biodistribution of drugs [194]. They act as drug vehicles and can target tumor cells or tissues while shielding the drug during transport [192]. The delivery of drugs to the site occurs actively, i.e., a drug delivery system (DDS) is coupled with peptides and antibodies anchored with lipids or receptors at the target site, or passively, i.e., the drug is transported via self-assembled nanostructured material and released at the target site [195].

Nanoparticles, in general, are excellent nanocarriers for targeted drug delivery. They serve as potential candidates due to their biocompatibility, controlled drug release, prolonged circulation time, and accumulation at the tumor site due to enhanced permeability and retention (EPR) effect [196,197,198]. Table 4 lists some common nanoparticles used for drug delivery, along with their advantages and disadvantages.

The use of QD nanoparticles for targeted drug delivery is of great interest due to their unique properties, including their distinctive optical characteristics due to their quantum confinement effects. QDs are also an excellent choice because of their intrinsic fluorescence and unique properties to serve as a multifunctional nanosystem. This includes their ability to aid in targeted drug delivery and improve the bioavailability and stability of drugs by prolonging the circulation time in vivo and improving distribution [216].

The use of QDs for drug delivery requires the modification of their surface with target ligands (e.g., thioglycolic acid, polyethylene glycol (PEG), antibodies, DNA, biotin, or peptides) [218,219]. Some surface modifications enable the drug molecules to bind to the QDs through covalent bonds or electrostatic binding, which forms nano-drug carriers and then makes fluorescent tags of drug molecules in cells and live animals [220]. Hence, QDs can act as drug carriers as well as fluorescent probes to trace drug distribution in vivo [221]. However, the size of the QD should be considered because excretion from the body is important. Moreover, the drugs can be loaded into a polymer NP system containing either hydrophilic QDs or hydrophobic QDs, depending on the polymer particle type used for encapsulation. This is followed by delivery at the desired site, where the polymer particle releases the drug via degradation at low pH or diffuses out of the polymer [222]. Figure 7 shows the development of molybdenum disulfide (MoS_2_) QDs for tumor fluorescence imaging, tumor targeting, and chemo/photodynamic therapy (PDT).

Table 5 shows in vitro and in vivo targeted drug delivery using QDs.

The anti-tumor drug Adriamycin was loaded into a drug delivery system (DDS) developed by Hao et al. [243] through covalent interactions and the formation of Zn^2+^-DOX. The lanthanum-doped zinc oxide (La-ZnO) QDs were modified with hyaluronic acid (HA). This enables them to bind specifically to receptor CD44. In addition, the developed system was PEGylated to stabilize it under physiological conditions. Their results showed that an anti-tumor effect and dual fluorescence enhancement were achieved due to lanthanum doping.

Similarly, Cai et al. [55] used covalent interactions and the formulation of a zinc doxorubicin (Zn^2+^-Dox) chelate complex to load Doxorubicin to hyaluronic-functionalized PEGylated zinc oxide (HA-ZnO-PEG). They reported that the system exhibited an acidic pH response, which triggered targeted drug release in tumors.

A polylactic acid (PLA) polymer matrix has been used for drug encapsulation as it provides sustained and controlled drug release. Gautam et al. [244] conjugated Gefitinib to polyethylene glycol graphene QDs (PEG-GQDs) and encapsulated the QDs in polylactic acid (PLA) microsphere for cancer therapy. They aimed to use the developed system for controlled drug (Gefitinib) delivery. They reported drug release to be around 65% after 48 h at an acidic pH (pH = 4.5). This was attributed to destabilized electrostatic interaction. At basic pH (pH = 7.4), drug release was observed to be slower. They suggested that their prepared system using PLA microspheres could be an excellent candidate for cell imaging and drug delivery. Figure 8 illustrates the in vitro release of Gefitinib-loaded microspheres.

Furthermore, Wei et al. [245] evaluated using QDs as an effective tool for microenvironment-targeted drug delivery. Using chemical oxidation and a covalent reaction, Pt-loaded and polyethylene glycol (PEG)-modified graphene QDs (GQDs) were developed as a drug delivery system. The Pt-loaded and PEG-GQDs were developed to overcome hypoxia-induced chemoresistance in oral squamous cell carcinoma. The accumulation of Pt within oral squamous cell carcinoma (OSCC) cells was significantly enhanced using polyethylene glycol–graphene QDs-Pt (GPt) in normoxia and hypoxia. The GPt was observed 2 h after incubation in the cytoplasm and in the nucleus 5–8 h after incubation. After 24 h, GPt luminescence was further enhanced, indicating that GQDs can transfer Pt and are potential platforms for nucleus-targeted drug delivery. The in vivo studies reported that GPt inhibited tumor growth.

In another study, graphene QDs (GQDs) were incorporated into carboxymethyl cellulose (CMC) hydrogels to design a hydrogel nanocomposite film loaded with doxorubicin as a drug model. They reported drug release to inversely depend on the concentration of GQD (i.e., release % of DOX from CMC/GQD decreases with increasing GQD concentration) even as the pH was varied. In addition, increasing GQD concentration resulted in increased drug loading capacity, showing that GQDs incorporated in CMC films resulted in pH sensitivity and the prolonged release of the therapeutic agent [246]. Olerile et al. [247] developed paclitaxel (PTX) and CdTe@CdS@ZnS QDs co-loaded in nanostructure lipid carriers (NLC). Their experiments showed that the encapsulation efficiency of PTX was 80.70 ± 2.11% and the drug loading was 4.68 ± 0.04%. In addition, the rate of tumor suppression was reported to be 77.85%. Their results showed that the co-loaded NLC could also detect H22 tumors, revealing some potential for bioimaging.

Zhao et al. [248] also used paclitaxel (PTX) as a model drug. They synthesized manganese-doped zinc selenide zinc sulfide (ZnSe:Mn/ZnS) core/shell, and the anti-cancer drug (PTX) was co-loaded into hybrid silica nanocapsules conjugated with folate. Folic acid (FA) conjugation was performed via an esterification reaction between FA carboxylic groups and animated F127 amino groups. The PTX solubility (0.1 µg/mL) was reported to be enhanced 630 times, improving the loading amount to 62.99 µg/mL. Their reports showed sustained release of PTX across 12 h. Overall, the developed hybrid nanocapsules showed the efficacy of anti-cancer drug loading and sustained release. Figure 9 illustrates the process of FA conjugation.

Demir Duman et al. [249] evaluated the use of near-infrared-emitting silver sulfide (Ag_2_S) QDs. The Ag_2_S QDs surfaces were coated with PEG, functionalized with Cetuximab (Cet) antibodies to target and reveal tumor cells, and loaded with the 5-fluorouracil (5FU) anti-cancer drug. The QDs were developed for targeted NIR imaging and treatment of lung cancer via low and high epidermal growth factor receptors (EGFR). The Cet-conjugated QDs delivered 5FU effectively and selectively to A549 cells and provided exceptionally enhanced cell death associated with apoptosis. They suggested their novel system would significantly overcome drug resistance compared to the treatment of 5FU alone.

Yang et al. [250] developed GQDs loaded into hollow mesoporous silica nanoparticles (HMSN cavity) (GQDs@hMSN-PEG NPs). The singlet oxygen (^1^O_2_) generating capacity of the GQDs was not affected after hMSN loading. The developed GQDs@hMSN-PEG NPs were reported to exhibit excellent absorption and emission properties. The drug loading capacity was measured and the NPs were found to carry significant amounts of DOX. They further demonstrated drug delivery feasibility on mice bearing 4T1 tumors by injecting GQDs@hMSN (DOX)-PEG, with results showing the feasibility of tumor-directed drug delivery. Table 6 summarizes some other applications relative to cancer involving QDs.

## 6. Cytotoxicity

The cytotoxicity of many QDs is a major deterrent to using QDs in widespread biomedical imaging and therapy. Despite their promising potential in various applications due to their optoelectronic properties, the toxicity of QDs limits their use to in vitro or animal studies. The toxicity of QDs is attributed to their chemical compositions containing heavy metal ions such as cadmium and indium [269]. In addition, their environmental conditions and physicochemical structure contribute to toxin availability (e.g., size, concentration, capping material, mechanical stability, etc.) [270,271,272]. For instance, the cardiotoxicity of cadmium selenide zinc sulfide (CdSe/ZnS) QDs was investigated by Li et al. [273]. A significant amount of cadmium (Cd) was detected in the hearts of mice bearing CdSe/ZnS QDs. Their results showed the accumulation of CdSe/ZnS QDs in the heart in addition to the incomplete QD excretion of up to 42 days.

In another study, the toxicity of copper indium disulfide zinc sulfide (CuInS_2_/ZnS) core/shell QDs was investigated in vivo. Ninety days after injection, indium was detected in the kidney, heart, brain, and testis. In another study, CuInS_2_/ZnS QDs were reported to accumulate in the liver and spleen [274].

Furthermore, QD toxicity results from the generation of reactive oxygen species (e.g., free radicals and the creation of singlet oxygen) [275], which could damage DNA. Near-infrared (NIR) QDs have also been reported to present a health risk. For instance, Zhang et al. [276] reported that lead sulfide/cadmium sulfide (PbS/CdS) QDs (0.7%) remained in mice after 1 month. The QDs were observed in the liver, spleen, lungs, kidneys, stomach, and gut and distributed to other body parts. The toxicity and accumulation of QDs in off-target tissues is an issue that must be addressed.

Conversely, researchers have reported that the coating of QDs or the surface functionalized QDs reduced the leaching of ions [277], thereby reducing acute toxicity. For instance, Murase et al. [278] synthesized cadmium selenide/zinc sulfide (CdSe/ZnS) QDs encapsulated in highly emitting silica capsules by the sol–gel method. At a shell thickness of 15 nm, the release was suppressed effectively compared to a shell thickness of 10 nm. They further reported leakage suppression at a temperature of 40 °C. Their results revealed that the silica capsules were non-toxic to cells. There is still the need to consider the effective surface coating of QDs because a better-protecting shell is less likely to leach heavy metals; however, at the same time, the size of QDs is increased after encapsulation, which might hinder their use in some applications. Even if capping can effectively minimize toxic ion release and preclude acute toxicity, the long-term buildup of capped QDs must be addressed for clinical translation to be approved. Consequently, additional investigation is warranted to develop improved methods for synthesizing QDs that mitigate or eradicate their toxic properties.

## 7. Conclusions

The use of nanoparticles in the fight against cancer has been researched extensively. Nanoparticles possess several characteristics required to overcome the limitations of conventional cancer management strategies, thus providing a platform for early detection and treatment. Quantum dots are the latest nanoparticles to exhibit unique properties that could impact how cancer is diagnosed and treated. These features include their small tuneable size, stable photoluminescence, large surface-to-volume ratio, and potential biocompatibility. QDs have been extensively applied for in vitro and in vivo tumor imaging and, more specifically, integrated with therapeutic agents for targeted drug delivery in vivo. The flexibility to bioconjugate or modify the surface of QDs according to the needed application qualifies QDs to be potential candidates as multifunctional systems. Many studies have shown that drug encapsulation in QDs increased drug delivery efficacy. More importantly, surface-modified QDs show promise as a great platform that could simultaneously deliver loaded drugs and provide real-time imaging of the biodistribution of the drug at tumor sites in vitro and in vivo.

Furthermore, studies have revealed that QDs subjected to surface modification serve as fluorescent markers and can inhibit tumor growth substantially or directly induce tumor cell death when combined with the requisite receptors or ligands. While the toxicity issues associated with QDs containing heavy metals like cadmium have been acknowledged, their tendency to accumulate in bodily organs due to their overall size hinders some of their potential use in human in vivo imaging and drug delivery applications. Hence, the development of heavy-metal-free QDs is extensively studied for possible clinical applications [279]. While acknowledging the need to minimize QD dimensions and appropriately capping them to mitigate toxicity, all while considering the specific application needs, it is important to note that QDs have demonstrated novel and useful promise in cancer imaging and treatment. Without a doubt, persistently utilizing the QD platform for cancer-related biological research will lead to a noteworthy breakthrough that has the potential to reshape the current research landscape.

## Figures and Tables

**Figure 1 nanomaterials-13-02566-f001:**
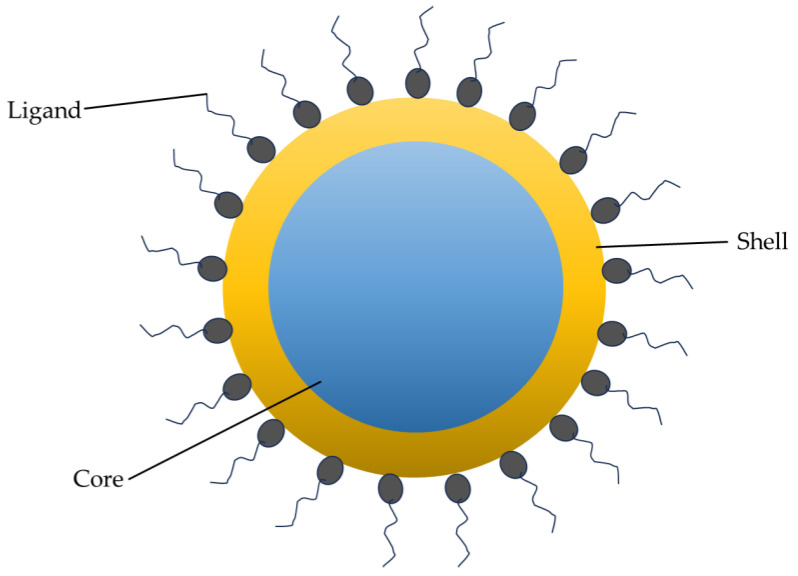
Structure of a QD showing the core/shell/ligand.

**Figure 2 nanomaterials-13-02566-f002:**
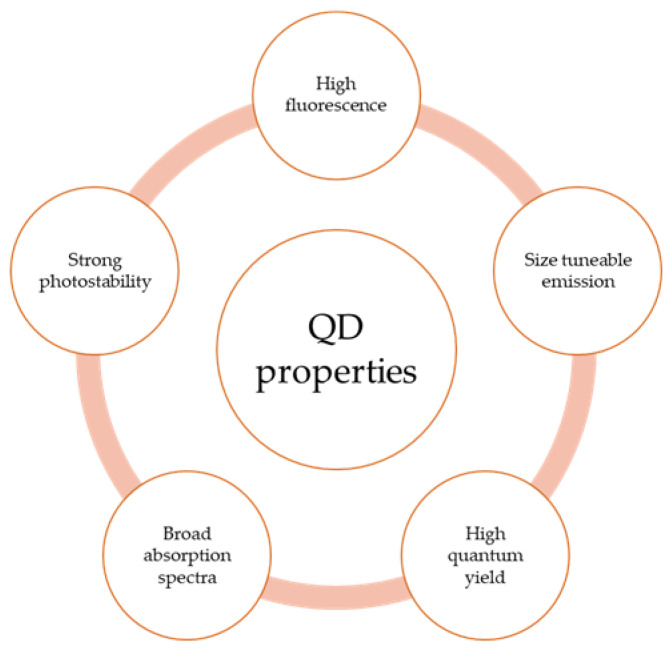
Diagram showing some optical properties of QDs.

**Figure 3 nanomaterials-13-02566-f003:**
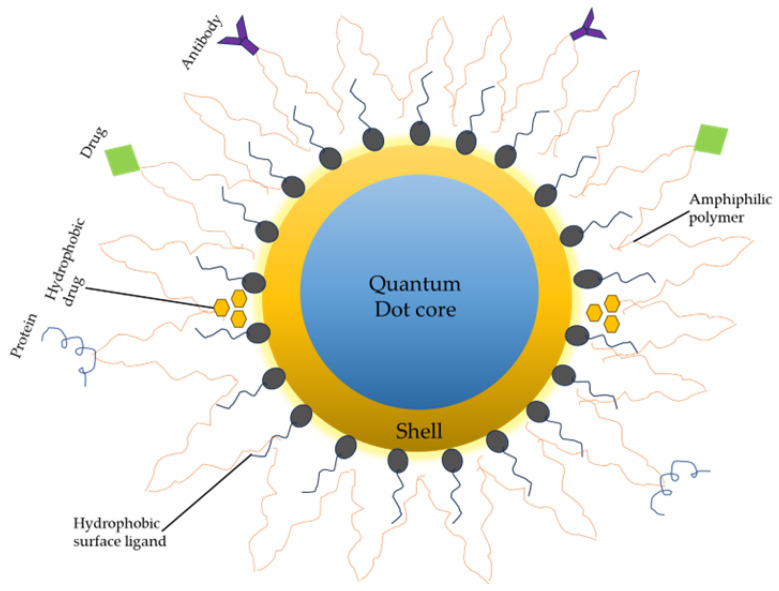
Surface functionalization of QD core/shell. The surface coating (e.g., amphiphilic polymer coating) enables antibodies, drugs, proteins, and other compounds to be linked with it. Hydrophobic drugs can also be integrated between the hydrophobic core and amphiphilic layer.

**Figure 4 nanomaterials-13-02566-f004:**
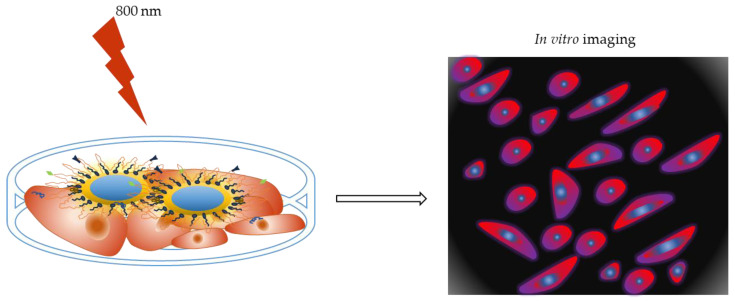
Schematic representation of QDs for in vitro tumor imaging.

**Figure 5 nanomaterials-13-02566-f005:**
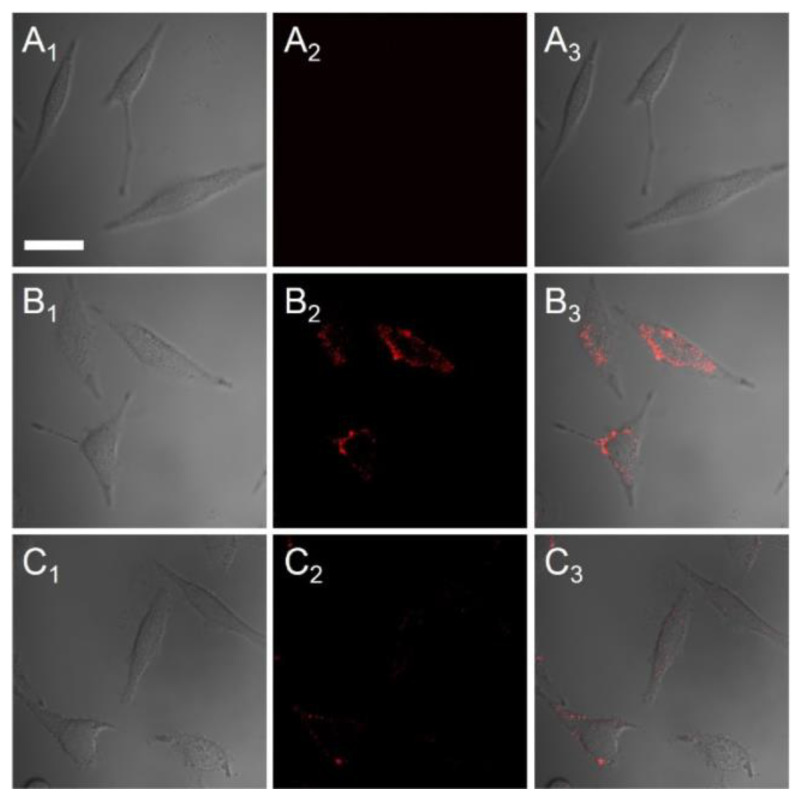
Confocal fluorescence images of HeLa cells (**A**) before and (**B**) after mixing with CdTe/CdS QDs at 5 µg/mL and (**C**) 30 µM of Cu^2+^ was then added to (**B**) to monitor concentration change of Cu^2+^ Showing as (1) brightfield images, (2) fluorescence images (700–800 nm filter), and (3) merging of (1) and (2). (Scale bar 30 µm. Reprinted with permission from [181]).

**Figure 6 nanomaterials-13-02566-f006:**
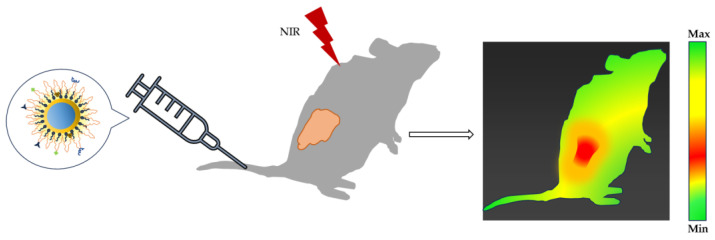
Schematic representation of QD injected into a tumor-bearing mouse for in vivo tumor imaging.

**Figure 7 nanomaterials-13-02566-f007:**
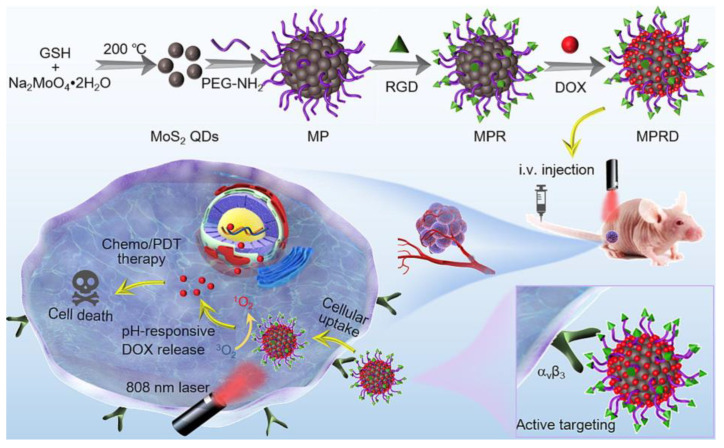
Illustration of synthesized PEGylated MoS_2_ conjugated with arginine-glycine-aspartic acid (RGD) peptide to form MPR (i.e., novel nanocarrier) and then loaded with doxorubicin (DOX) to form MPRD. MPRD exhibits tumor-targeting ability, pH-responsive drug release, and synergistic chemo/PDT performance under near-infrared (NIR) laser irradiation(grey circles: synthesized MoS_2_ QDs, red circles: Dox, green triangles: RGD). Reprinted with permission from [223].

**Figure 8 nanomaterials-13-02566-f008:**
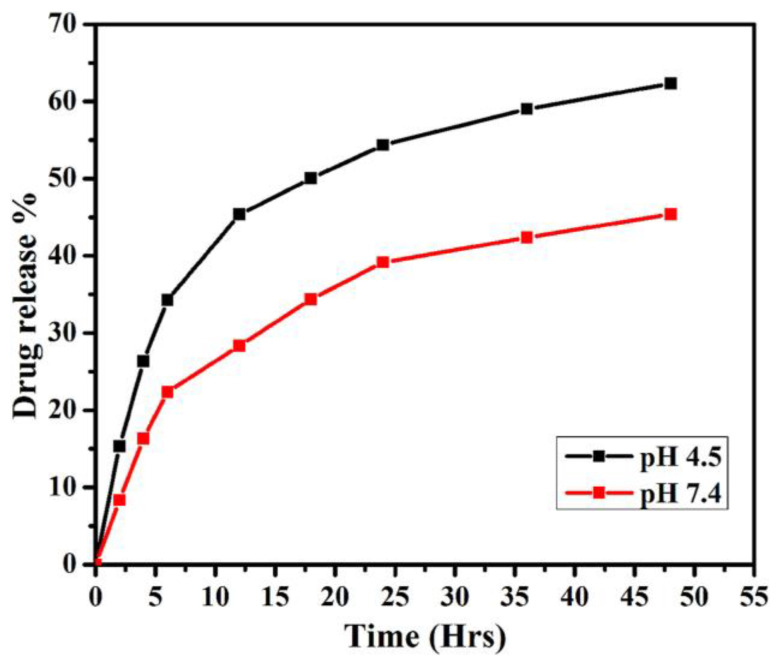
In vitro release of drug (Gefitinib)-loaded microspheres at pH 4.5 and 7.4. Reprinted with permission from [244].

**Figure 9 nanomaterials-13-02566-f009:**
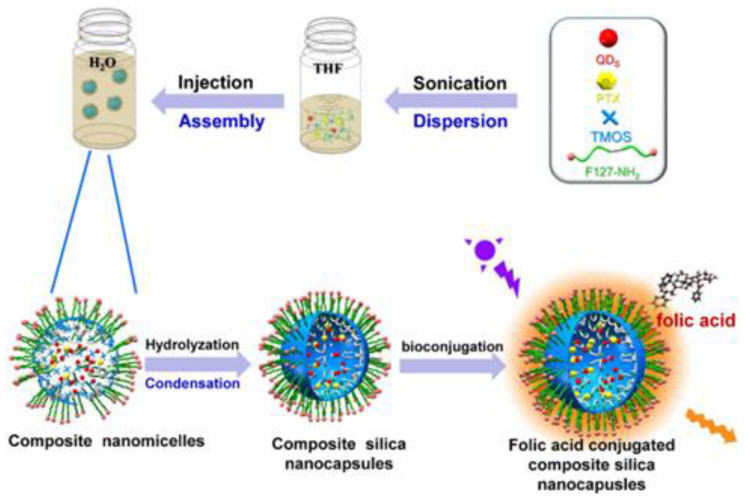
Schematic diagram of FA-conjugated hybrid silica nanocapsules. Reprinted with permission from [248].

**Table 1 nanomaterials-13-02566-t001:** Size of QDs and their emission wavelengths. Reprinted with permission from [64].

Quantum Dots	Size Range (Diameter nm)	Emission Range (nm)
Cadmium sulfide (CdS)	2.8–5.4	410–460
Cadmium telluride (CdTe)	3.1–9.1	520–750
Cadmium selenide (CdSe)	2–8	480–680
CdTe/CdSe	4–9.2	650–840
Indium phosphide (InP)	2.5–4.5	610–710
Indium arsenide (InAs)	3.2–6	860–1270
Lead selenide (PbSe)	3.2–4.1	1110–1310
1-Dodecanethiol silver sulfide ((Dt)-Ag_2_S)	5.4–10	1000–1300

**Table 2 nanomaterials-13-02566-t002:** The different synthesis techniques used to fabricate QDs.

Synthesis Methods	QDs Fabricated	Properties	Refs.
Electron beam lithography	QD nanostructures	Optical properties retained after cross-linking	[67]
	QD microarrays	Fluorescence Bioaffinity	[68]
Reactive ion etching	Indium gallium nitride (InGaN) QDs	Strong and distinct photoluminescence signal	[71]
Sol-gel	Titanium dioxide (TiO_2_) QDs	large surface area photocatalytic properties	[80]
	Zinc selenide (ZnSe) QDs embedded in Silicon dioxide (SiO_2_)	-	[81]
	Cadmium sulfide (CdS) and Ni-doped CdS	Highly crystalline	[113]
	Zinc oxide (ZnO)@polymer core/shell	Quantum yield above 50%	[114]
	Zinc oxide (ZnO) QD	High photoluminescence quantum yield	[115]
Microemulsion (reverse micelle)	Zinc sulfide (ZnS) QDs	Pure nanocrystal Quantum confinement effect Photoluminescence peak at 365 nm	[84]
	Cadmium sulfide/Zinc sulfide (CdS/ZnS) semiconductor QDs	Excellent luminescence and photostability	[86]
	Cadmium selenide@Zinc sulfide (CdSe@ZnS) within monodisperse silica	Good monodispersity High luminescence	[89]
Microemulsion (gas contacting technique)	Zinc selenide (ZnSe) QDs	Excellent photostability and size-dependent luminescence	[85]
Microemulsion method + ultrasonic waves (sono-microemulsion method)	Cadmium sulfide (CdS)	Narrow size distribution High crystallinity and purity	[116]
Physical vapor deposition	Niobium pentoxide (Nb_2_O_5_) QDs	Quantum confinement effect	[93]
RF magnetron sputtering	Cadmium selenide (CdSe) QDs	Optical properties	[100]
Solvothermal	Zinc Oxide (ZO) QDs	Small size Pure, high crystallinity and surface area	[117]
	Graphene QDs (GQDs)	11.4% photoluminescence quantum yield High stability Biocompatibility Low toxicity	[118]
Hydrothermal	Nitrogen- and sulfur-doped carbon QDs (N, S-doped CQDs)	Small Spherical Green emission	[119]
		Fluorescence quantum yield (10.35%)	
	Nitrogen-doped carbon QDs (N-CQDs)	Low toxicity Good photostability	[102]
	Silicon QDs	Good water dispersibility Strong photoluminescence High pH stability	[120]
	Tin oxide/Tin sulfide in reduced bovine serum albumin (SnO_2_/SnS_2_ @r-BSA2)	Specific selectivity Long term stability Enhanced reproducibility	[121]
	Nitrogen-doped Graphene QDs (N-GQDs)	High quantum yield Long-term fluorescence stability High sensitivity and specificity	[122,123]
Molecular beam epitaxy	Indium arsenide gallium arsenide core/shell (InAs/GaAs) QDs	Strong photoluminescence intensity High structural properties	[124]

**Table 3 nanomaterials-13-02566-t003:** A summary of surface functionalization of QDs (showing the advantages and disadvantages of the four main techniques).

Surface Modification Techniques	Advantages	Disadvantages	Refs.
Ligand exchange	Ease of processing Small QD size	Degradation of QD photophysical properties in an aqueous environment (i.e., reduced PLQY) QD core is susceptible to oxidation	[51,168,169,170]
Surface silanization	Improves biocompatibility Highly cross-linked ligand molecules End terminal groups allow further coating through the exposure of the terminal ends (e.g., thiol). Control of silica shell thickness encourages fine-tuning of QD response to light. Improves PLQY of QDs Improves photochemical stability	Large hydrodynamic size Aggregation of QDs in aqueous solution	[171,172,173]
Amphiphilic ligands	More chemically stable Increased colloidal stability Good biocompatibility and strong, stable fluorescence signals	Size enlargementSurface defects	[138,174,175]
Microsphere coating	Improve QD stability High fluorescence Can mask QD toxicity effectively	The formation of a uniform microsphere is hindered. Reduced PLQY Encapsulation of high concentrations of QDs results in QD aggregation	[167,176,177]

**Table 4 nanomaterials-13-02566-t004:** Advantages and disadvantages of organic and inorganic NPs used for drug delivery.

Organic Nanoparticles	Advantages	Disadvantages	Refs.
Liposomes	Enhances drug solubility	Decreased stability	[199,200]
	Reduces drug toxicity		
Micelles	Improves circulation time	Lack of targeting moieties	[201,202]
	Protects aqueous drug cargo		
Polymer NP (Chitosan)	Increase drug residence time in the bloodstream	Initial burst release results in loss of drug efficiency	[203,204]
Dendrimers	The hydrophobic core allows insoluble anti-tumor drugs to be absorbed and provides smooth delivery.	Rapid clearance of reticuloendothelial system	[205,206]
	The hydrophilic part increases stability and limits the particles' interaction with serum proteins.		
**Inorganic Nanoparticles**			
Silver NPs	Enhances PTX distribution in tumor microenvironment	Release of silver ions in cytosol	[207,208]
Gold NPs	Enhances photothermal therapy	Low tissue clearance	[209,210,211]
	Easily functionalized		
Mesoporous silica NPs	Controlled drug release	Slow biodegradation	[204,212]
Magnetic NPs (iron oxide)	Precise targeting of cancer cells Release of PTX under external magnetic field	Removal by macrophages	[213,214,215]
Quantum Dots	Improves the bioavailability of the drug	Leaching of heavy metals	[216,217]

**Table 5 nanomaterials-13-02566-t005:** In vitro and in vivo targeted drug delivery using QDs.

QDs Used In Vitro	Drug	Cell Line	Ref.
Iron oxide carbon QDs encapsulated in chitosan (Fe_2_O_3_/CQDs/Chitosan)	Curcumin	MCF-7 cells	[224]
Transferrin (TF)-conjugated Carbon QDs	Doxorubicin	MCF-7 cells	[225]
Graphene oxide QDs conjugated with glucosamine and boric acid (GOQDs-GlcN-BA)	Doxorubicin	MCF-7 cells	[226]
Magnesium nitride (Mg/N) doped carbon QDs (CQDs)	Epirubicin (EPI)	4T1 and MCF-7 cells	[227]
Nitrogen-doped Graphene QDs (N-GQDs)	Methotrexate (MTX)	MCF-7 human breast cancer cells	[228]
PEGylated molybdenum disulfide QDs (PEG-MoS_2_ QDs)	Doxorubicin	U251 cells	[229]
Zinc oxide adipic dihydrazide heparin (ZnO-ADH-Hep)	Paclitaxel	A549 cells	[230]
Cadmium-sulfide-modified chitosan (CdS@CTS)	Sesamol	MCF-7 cell	[231]
PEGylated Silver graphene QDs (Ag-GQDs)	Doxorubicin	HeLa and DU145 cells	[232]
Magnetic carbon triazine dendrimer reacted with graphene QDs (Fe_3_O_4_@C@TD GQDs) microsphere	Doxorubicin	A549 cell	[233]
**QDs used in vivo**			
Graphene QDs	Doxorubicin	MCF-7 cells	[234]
Silver sulfide (Ag_2_S) QDs conjugated with chitosan	Doxorubicin	HeLa cells	[235]
Manganese doped zinc sulfide (Mn-ZnS) QDs conjugated with folic acid (FA)	5-fluorouracil (5-FU)	4T1 breast cancer cells	[236]
PEGylated silver sulfide Ag_2_S QDs	Doxorubicin	MDA-MB-231 human breast tumor cells	[237]
Graphene QD (GQD)-modified magnetic chitosan Fe_3_O_4_@CS	Doxorubicin	Hepatocellular carcinoma	[238]
Red-emissive carbon QDs (CQDs)	Doxorubicin	HeLa cells	[239]
Black phosphorus QDs (BPQDs) encapsulated in platelet-osteosarcoma hybrid membrane (OPM)	Doxorubicin	Osteosarcoma	[240]
Nitrogen-doped carbon QDs conjugated with folic acid (FA)	Doxorubicin	4T1 and MCF-7 cells	[241]
PEGylated molybdenum disulfide (MoS_2_) QDs conjugated with arginylglycylaspartic acid (RGD) peptide	Doxorubicin	HepG2 cells	[223]
Polyethyleneimine (PEI)-conjugated graphene QDs (GQDs)	Doxorubicin	HCT116 cells	[242]

**Table 6 nanomaterials-13-02566-t006:** Other recent applications of QDs relative to cancer [226,251,252,253,254,255,256,257,258,259,260,261,262,263,264,265,266,267,268].

QDs Utilized	Application	Target Cells
Carbon QDs (CQDs)	Drug delivery	Breast cancer cell line
Carbon QDs (CQDs)	Drug delivery	Breast MCF-7 cancer cells
Graphene QDs (GQDs)	Drug delivery	U251 glioma cells
Near-infrared (NIR) copper indium sulfide zinc sulfide core/shell (CuInS_2_/ZnS) QDs	In vivo	RR1022 Cancer cell
Alloyed Zinc copper indium sulfide (ZCIS) QDs	In vitro	HER2-positive SKBR3 cancer cells
Molybdenum disulfide (MoS_2_) QDs-MXene	Electrochemiluminescence (ECL) sensor for detection	Gastric cancer cell exosome
Zinc oxide (ZnO) QDs	Drug delivery	HepG2 cells
Molybdenum disulfide (MoS_2_) QDs	Photodynamic therapy Drug delivery	HeLa and HepG2 cells
Manganese-doped molybdenum disulfide (Mn-MoS_2_) QDs	In vivo MR imaging Fluorescence labeling	786-O Renal carcinoma cells
Titanium-ligand-coordinated black phosphorus QDs (TiL4@BPQDs)	In vivo Photoacoustic Imaging	MCF-7 cancer cells
Graphene QDs (GQDs)	Photothermal therapy	MDA-MB-231
Folic-acid-conjugated carbon QDs (FA-CQDs)	Fluorescence imaging	MCF-7 cells and ovarian cancer (HeLa)
Copper indium sulfide zinc sulfide core/shell (CuInS/ZnS) QDs	Sensor probe for targeted imaging	BEL-7402 cancer cells
Titanium nitride (Ti_2_N) QDs	Photoacoustic (PA) imaging-guided photothermal therapy (PTT) in near-infrared (NIR-I/II) biowindows	293T, 4T1 and U87 cancer cells
Cadmium telluride cadmium sulfide (CdTe/CdS) core–shell QDs	Fluorescence imaging	MDA-MB-231/MDR
Zinc oxide (ZnO) QDs	Drug delivery	MCF-7
Cadmium selenide telluride zinc sulfide (CdSeTe/ZnS) QDs	Photothermal therapy	Hepatoma cells Huh7
Graphene QDs (GQDs)	Drug delivery	MCF-7 cells
Near-infrared (NIR) silver selenide (Ag_2_Se) QDs	In vivo tumor imaging	MCF-7 human breast cancer cells and SW1990 pancreatic cancer cells

## Data Availability

Not applicable.

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
