# Peer review of "Recent Breakthroughs in Using Quantum Dots for Cancer Imaging and Drug Delivery Purposes"

_nanomaterials, 2023, doi:10.3390/nano13182566_

Round 1

Reviewer 1 Report

COMMENTS TO AUTHORS:

Recommendation: Publish after minor revisions.

In this review, the authors present an intriguing insight into the field of quantum dots (QDs), a family of semiconductor nanoparticles with dimensions on the nanometer scale. This nanotechnology has recently found success in a variety of electronic and biological industries, with a variety of applications. Following a brief introduction to QDs, the authors also provide a summary of the two main synthetic approaches for the synthesis of QDs, their functionalization, and their application in cancer therapy. Overall, the work is interesting and written in an accessible manner. I would recommend this manuscript for publication in Nanomaterials, with the following suggestions:

1. The title: It would be helpful to highlight in the title that the work focused on the most recent breakthroughs in using QDs for in vitro and in vivo imaging and target drug delivery platforms in cancer therapy, emphasizing the work's novel nature.

2. There are several grammatical problems as well. Please proofread and edit.

3. The study's contribution to the field at the conclusion of the introduction should be improved.

4. It would have been interesting to obtain some information about the potential environmental impact of using these QDs, particularly in aquatic ecosystems (i.e. as references)

 There are several grammatical problems

Author Response

Please find the attached rebuttal for all three reviewers.

Reviewer 2 Report

This work impresses with the amount of analyzed information and clear systematization.

The review very fully describes the current state of research in the field of synthesis, modification, applications and properties of quantum dots.

The only thing that could be added is a graphic abstract, because the visual component could be improved.

Author Response

(The authors gave the same response as above.)

Reviewer 3 Report

The present review is an exhaustive work about QDs, well organized and making easy to the reader follow the story. The sections are carefully presented and clear, the number of cites is correct, and the objective of the paper is properly explained in the introduction. I reccomend publication in present form, after minor format corrections:

- Modify the reference format to make them more homogeneous. In some points they are refered like a range (i.e. [18]-[22]) and in other places as an enumeration (i.e. [30], [31], [32], [33]). I suggest the [18-22] format.

- Line 50, it is more correct to say that in nanomaterials, the size of AT LEAST ONE OF THEIR DIMENSIONS is between 1-100 nm.

- If possible, introduce more figures in section 5 to help reader to follow even more the evolution from general statements to specific aplications. They can be drawn by authors or simply cite with the right permissions the most interesting ones.

Author Response

(The authors gave the same response as above.)
